# Effect of Enzymatic, Ultrasound, and Reflux Extraction Pretreatments on the Chemical Composition of Essential Oils

**DOI:** 10.3390/molecules25204818

**Published:** 2020-10-20

**Authors:** Anđela Miljanović, Ana Bielen, Dorotea Grbin, Zvonimir Marijanović, Martina Andlar, Tonči Rezić, Sunčica Roca, Igor Jerković, Dražen Vikić-Topić, Maja Dent

**Affiliations:** 1Faculty of Food Technology and Biotechnology, University of Zagreb, Pierottijeva 6, 10 000 Zagreb, Croatia; amiljanovic@pbf.hr (A.M.); dorotea.polo@gmail.com (D.G.); martina.andlar@gmail.com (M.A.); trezic@pbf.hr (T.R.); 2Faculty of Chemistry and Technology, University of Split, Ruđera Boškovića 35, 21 000 Split, Croatia; zmarijanovic@ktf-split.hr (Z.M.); igor@ktf-split.hr (I.J.); 3NMR Centre, Ruđer Bošković Institute, Bijenička cesta 54, 10 000 Zagreb, Croatia; sroca@irb.hr (S.R.); dvikic@gmail.com (D.V.-T.); 4Department of Natural and Health Sciences, Juraj Dobrila University of Pula, Zagrebačka 30, 52 100 Pula, Croatia

**Keywords:** Clevenger hydrodistillation, *Salvia officinalis* L., *Rosmarinus officinalis* L., *Laurus nobilis* L., cell wall-degrading enzymes, ultrasound pretreatment, reflux extraction pretreatment, GC-MS analysis

## Abstract

The effect of different hydrodistillation pretreatments, namely, reflux extraction, reflux extraction with the addition of cell wall-degrading enzymes, and ultrasound, on the yield and chemical composition of essential oils of sage, bay laurel, and rosemary was examined. All pretreatments improved essential oil yield compared to no-pretreatment control (40–64% yield increase), while the oil quality remained mostly unchanged (as shown by statistical analysis of GC-MS results). However, enzyme-assisted reflux extraction pretreatment did not significantly outperform reflux extraction (no-enzyme control), suggesting that the observed yield increase was mostly a consequence of reflux extraction and enzymatic activity had only a minute effect. Thus, we show that ultrasound and reflux extraction pretreatments are beneficial in the production of essential oils of selected Mediterranean plants, but the application of enzymes has to be carefully re-evaluated.

## 1. Introduction

Bioactive compounds are secondary plant metabolites with potential for many applications in human use. Due to the high content of bioactive compounds in essential oils [1,2,3] and their ability to stop or delay the aerobic oxidation of organic matter, their application in the food industry, cosmetic industry, phytotherapy, medicine, and many other fields is constantly growing [4,5,6,7,8].

Essential oils are usually isolated by steam- or hydrodistillation. Although these methods have been successfully used for years, novel extraction procedures are being developed in recent decades, with the aim to increase the yield of essential oils and/or improve their chemical composition, while shortening the extraction time [9]. To get closer to that ideal goal, novel production procedures, like ultrasound or enzymatic degradation of the cell wall, are being applied as pretreatments to distillation. Benefits of ultrasound application in solid–liquid extraction are the intensification of mass transfer, improved solvent penetration into the plant tissue, and capillary effects. The collapse of cavitation bubbles near the cell walls is assumed to cause the cell disruption and at the same time a good penetration of the solvent into the cells through the ultrasonic jet [10]. Several studies have reported that the addition of ultrasonic pretreatment before hydrodistillation leads to a shortening of the extraction time [11,12,13,14], sometimes also accompanied by yield increase [12,13,15,16,17] and improvement in oil chemical composition [9,12,13,14].

Further, enzyme-assisted extraction has been intensively studied in the last decade, since the plant cell wall, as a structure composed mainly of resistant polymers like cellulose, xylan, lignin, and pectin, can reduce the extraction efficiency. Enzymes, like cellulase, xylanase, and pectinase, can degrade or disrupt cell wall components, thus enabling better release and more efficient extraction of bioactive compounds from plants and improving the bioactive content of essential oils and extracts [2,18,19,20,21,22]. However, while some authors reported that enzyme-assisted extraction enhanced essential oil yield and improved their chemical composition [2,19,21,22,23,24,25,26], others did not demonstrate such effect [1], indicating that further research of enzyme application in plant extractions is necessary.

We hypothesized that different hydrodistillation pretreatments, namely, ultrasound-assisted extraction, reflux extraction, and enzyme-assisted reflux extraction, can improve the yield of essential oils of selected Mediterranean wild plants: sage (*Salvia officinalis* L.), rosemary (*Rosmarinus officinalis* L.), and bay laurel (*Laurus nobilis* L.). Also, we hypothesized that these relatively mild pretreatments will not adversely affect the chemical composition of essential oils.

## 2. Results

### 2.1. Cellulase, Xylanase, and Pectinase Can Degrade Their Respective Substrates under the Reaction Conditions

Before applying cell wall-degrading enzymes (xylanase, pectinase, and cellulase) for the extraction of bioactive compounds from the plant material, their activity under the extraction conditions (1 h, 40 °C, in MiliQ water) was measured at a small scale by 3,5-dinitrosalicylic acid (DNSA) assay. The enzymes were mixed with their appropriate polymeric substrates in the MiliQ water and the reaction was incubated for 1 h at 40 °C. The results have shown that all enzymes were able to degrade their substrates: xylanase degraded xylan to 1.10 U/mL, pectinase pectin to 0.03 U/mL, and cellulase cellulose to 0.05 U/mL.

### 2.2. Optimization of Ultrasound Extraction Conditions

Optimal ultrasonic power and duration, chosen based on the overall maximum total phenol yield, were 30% of maximal ultrasonic power during 10 min (Table 1). In some cases, total phenol yield was slightly higher after 15 min ultrasound treatment (e.g. at 30% of maximal ultrasonic power for rosemary, and 90% for bay laurel). Still, since 15 min ultrasound treatment caused intensive solvent evaporation, the shorter time of 10 min was chosen as optimal.

### 2.3. Different Pretreatments Increased the Essential Oil Yield

All tested hydrodistillation pretreatments caused an increase in essential oil yield when compared to no-pretreatment control (HD): 56–64% for bay laurel, 40–60% for rosemary, and 45–55% for sage (Figure 1). However, only slight differences in essential oil yield were observed between different pretreatments for the same plant species. Importantly, none of the individual (HD-REX, HD-REC, HD-REP) or combined (HD-REPCX) enzymatic pretreatments significantly outperformed reflux extraction pretreatment (HD-RE). Since HD-RE served as a no-enzyme control, this indicates that enzymatic activity had only a minute effect on essential oil yield increase and that it should be mostly attributed to reflux extraction *per se*.

When comparing different plants, Clevenger hydrodistillation of bay laurel leaves consistently resulted in the highest oil yield (from 1.95 mL oil/100 g dry plant for HD-REP to 2.05 mL oil/100 g dry plant for HD-REX and HD-REPCX), followed by sage (from 1.45 mL oil/100 g dry plant for HD-RE to 1.55 mL oil/100 g dry plant for HD-REX and HD-REPCX) and rosemary (from 0.7 mL oil/100 g dry plant for HD-REC to 0.8 mL oil/100 g dry plant for HD-RE).

### 2.4. Overall Chemical Composition of Essential Oils Was Not Significantly Affected by Different Hydrodistillation Pretreatments

The chemical composition of obtained essential oils was analyzed by GC-MS (Appendix A, Figure 2) and NMR (Supplementary file 1, Appendix A). The pretreatments did not significantly affect the quality of essential oils (*p* < 0.05), that is, the oil composition was comparable to the no-pretreatment control, as confirmed by Spearman’s test performed using GC-MS data (Supplementary file 2, Appendix A). GC-MS analysis showed that the essential oils of all plants were richest in monoterpenes: up to 74%, 71%, and 62% of total peak area in sage, bay laurel, and rosemary essential oils, respectively (Appendix A, Figure 2). Sesquiterpenes were also well represented: up to 35%, 17%, and 56% of total peak area in sage, bay laurel, and rosemary essential oils, respectively. Dominant compounds (i.e., terpenes represented with more than 5% of total peak area) were confirmed by NMR (as described and represented in Supplementary file 1, Appendix A and Appendix 1): linalool, camphor, borneol, and berbenone in rosemary essential oil (Appendix AA), camphor, manool, α- and β-thujone, and veridiflorol in sage essential oil (Appendix AB), and α-terpenyl acetate, linalool, and methyleugenol in bay laurel essential oil (Appendix AC).

Seventy-one components were identified in the sage essential oils by GC-MS, representing up to 97% of total GC peak areas (Appendix A). Sage essential oils were richest in oxygenated monoterpenes, such as α- and β–thujone (up to 23% and 10%, respectively), camphor (up to 17%), borneol (up to 8%), and 1,8-cineole (up to 8%). Additionally, some sesquiterpenes were present in significant quantities, such as manool and veridiflorol (up to 14% each). Among other compounds, phenylpropane derivates, eugenol (up to 3%), and methyleugenol (up to 2%) were detected with significant percentages, although much lower than monoterpene and sesquiterpene major compounds.

In the bay laurel essential oils, GC-MS allowed the identification of 84 components, covering up to 99% of the total GC profile (Appendix A). Major detected monoterpenes were α-terpenyl acetate (up to 18%), 1,8-cineole (up to 27%), and linalool (up to 8%). Among sesquiterpenes, the most abundant were veridiflorol (up to 4%), trans-caryophyllene (up to 4%), bicyclogermacrene (up to 1.4%), and β-elemene (up to 1.6%). Other compounds detected in significant quantities were phenylpropane derivates, methyleugenol (up to 10%), and eugenol (up to 9%).

Finally, 60 components were detected in the rosemary essential oil, covering up to 95% of total GC peak areas (Appendix A). Rosemary essential oil was richest in oxygenated monoterpenes, such as borneol (up to 24%), camphor (up to 18%), linalool (up to 6%), and 1,8–cineole (up to 9%). Major sesquiterpenes were berbenone (up to 22%) and t-muurolol (up to 6%). Other compounds, like eugenol (up to 6%) and methyleugenol (up to 4%), were also present in significant amounts.

## 3. Discussion

We have tested the effect of different pretreatments using selected Mediterranean plants: rosemary, sage, and bay laurel, and demonstrated that different hydrodistillation pretreatments can improve the yield of respective essential oils. At the same time, the pretreatments did not significantly affect the quality of oils and their composition was comparable to no-pretreatment controls. For the first time, we have demonstrated that reflux extraction (e.g., soaking of milled plant material at 40 °C for one hour) could significantly improve essential oil yield.

Rosemary and sage are aromatic, medicinal plants that have received particular attention in the Lamiaceae family due to their aromatic and chemical composition [2,27,28], while bay laurel is a valuable medicinal plant from the Lauraceae family, which has been widely used as a spice and flavoring agent [19]. Due to the numerous bioactive compounds, the essential oils of these plants show a wide range of biological activities such as antimicrobial, preservative, antioxidant, and antifungal, which makes them valuable in a range of applications, from medicinal to the food industry [29,30,31,32,33]. Considering these valuable properties, the improvement of current and the development of novel essential oil extraction procedures is a subject of many recent studies [28,34,35].

In this study, we have obtained similar volatile profiles of sage, rosemary, and bay laurel essential oils as reported elsewhere [2,3,19,35,36,37,38,39]. Also, the essential oil composition obtained after different pretreatments was comparable to the essential oils obtained by direct hydrodistillation. Other studies had also shown that enzymatic and ultrasonic pretreatments did not affect the overall composition of the oil [11,15,16,17,21,22,26], although some authors reported that the quantities of individual major components varied significantly in relation to the extraction technique used [1,2,12,13,14,19,24]. For instance, Boulila et al. [19] reported that enzyme pretreatment of bay laurel leaves resulted in increased concentration of oxygenated monoterpenes, and explained such effect by a possible increase in oxidation after cell wall disruption or the presence of oxidases in the enzyme preparation. Also, cellulase pretreatment reportedly increased the amount of cis-verbenol and camphor and decreased the amounts of 1,8-cineol, α-pinene, fenchene, and terpinen-4-ol in rosemary essential oil [2]. We have also observed that the ratios of some components varied among different pretreatments. For example, in sage essential oil, the amount of almost all monoterpene hydrocarbons increased after HD-REPCX. On the other hand, HD-US resulted in a higher amount of major oxygenated monoterpenes (1,8–cineole, α-thujone, camphor). In bay laurel essential oil, all pretreatments, except HD-REX, resulted in an increase of 1,8-cineole, which is in line with the findings of Boulila et al. [19]. In rosemary essential oil, an increase of monoterpene hydrocarbons was noticeable after HD-US, while HD-RE resulted in an increase of 1,8-cineole and linalool, as well as phenylpropane derivatives eugenol and methyleugenol. However, such slight changes in the quantity of individual components observed after different pretreatments were not significant and did not change the overall oil quality, as confirmed by the statistical analysis of GC-MS results.

Among the different pretreatments applied in this study, the simplest was reflux extraction (HD-RE). The incubation of finely milled plant material for 1 h at 40 °C in MiliQ water before hydrodistillation resulted in up to 60% essential oil yield increase. It was previously shown that such soaking of the plant material in water [40] or acidic medium [1] before distillation may increase the quantity of oil by enhancing the leaching of ingredients from the already disrupted cells. Also, swelling and hydration of plant material, which enlarge the pores in cell walls and increase the turgor pressure in the still intact plant cells, might lead to the enhanced diffusion of the oil ingredients into the soaking medium. For example, soaking of agarwood in lactic acid for 168 h [41] and soaking of thyme leaves in distilled water overnight at 50 °C improved the essential oil yield [40]. Thus, soaking of plant material before hydrodistillation, as applied here for selected Mediterranean plants, presents a simple and cost-effective treatment that results in a significant increase in essential oil yield.

In addition, we have tested whether the application of cell wall-degrading enzymes prior to hydrodistillation has a positive effect on the extraction efficiency. We have followed previously described protocols [2,19] and applied separate and combined pretreatments with cellulase, pectinase, and xylanase to plant material. Prior to the extractions, we used a small-scale enzyme assay and confirmed the activity of all enzymes under the reaction conditions. However, the application of enzymes did not result in a significant increase in the yield of the essential oil above the reflux extraction pretreatments, suggesting that, in our case, the enzymatic activity had only a minor effect on the yield. On the contrary, the positive impact of enzyme application on the yield of essential oils was reported in multiple recent studies [2,19,21,22,23,25,26]. For instance, enzyme-assisted extraction pretreatment was used prior to hydrodistillation of bay laurel [19], rosemary, and thyme leaves [2], and reportedly resulted in an increase in essential oil yield (up to 109% for thyme leaves). However, these studies were lacking a no-enzyme control, and the observed yield increase was calculated by comparison with no-pretreatment control. In our case, the extraction results were compared to no-pretreatment control (HD) as well as to reflux extraction pretreatment (HD-RE), that served as a no-enzyme control since it was performed in the same conditions as the enzyme-assisted extraction pretreatments (MiliQ water at 40 °C for 1 h), only without addition of enzyme(s). In comparison to no-pretreatment control, an increase in essential oil yield was significant for all enzyme pretreatments individually, as well as for their combination, and there were no significant differences between each individual pretreatment. Unexpectedly, reflux extraction resulted in approximately the same essential oil increase as the enzyme-assisted pretreatments, leading to the conclusion that essential oil yield increase can be more attributed to reflux extraction pretreatment *per se* (i.e., soaking of macerated plant material in warm water) than to enzymatic degradation of the cell wall. The positive effect of the soaking of plant material alone, as observed here, could not be properly assessed and compared with the effect of enzymes in the previous studies that were lacking no-enzyme controls [2,19,21,22,23,24,25,26]. Our results are congruent with the results of Dimaki and coworkers [1] who analyzed the effect of enzyme pretreatments on hydrodistillation and ultrasound-assisted extraction of *Sideritis* sp. essential oil. They have used both no-pretreatment and no-enzyme control (i.e., preincubation in acidic medium) and also found that enzymatic pretreatment was not superior to the mere soaking of plant material in an acidic buffer. Also, acidic and enzymatic pretreatments prior to the ultrasound-assisted extraction resulted in a similar increase in extraction yield in comparison to no-pretreatment control. In conclusion, our results and results of Dimaki et al. [1] point out that the application of enzymatic pretreatments with the aim to increase the yield of essential oils, although often reported, should be carefully re-evaluated.

Finally, we have demonstrated the beneficial effect of ultrasound pretreatment on the overall extraction efficiency. In comparison to the control experiment, ultrasound-assisted extraction pretreatment caused a 50% improvement in sage and rosemary and 60% in bay laurel essential oil yield, similar to the results of reflux-assisted extraction. This is comparable with the results of other studies [12,13,15,16,17]. For instance, 30 min-ultrasonic-maceration pretreatment prior to the steam distillation of milled *Thymus vulgaris* L., *Mentha piperita* L., and *Origanum majorana* L. leaves resulted in approximately 10% essential oil yield increase, compared to no-pretreatment control [15,16]. Ultrasound-assisted extraction as a pretreatment to hydrodistillation of grounded carrot seeds increased essential oil yield by approximately 33% [17]. Hydrodistillation of grounded *Elettaria cardamomum* L. seeds with ultrasonic pretreatment resulted in a 4.9% essential oil yield increase and extraction lasted less than an hour, compared to control, which lasted 6 h [13]. Even when there was no yield increase, the application of ultrasound resulted in a significant shortening of the essential oil isolation procedure [11,14]. In conclusion, we report a high increase in essential oil yield after ultrasound pretreatment when compared to other studies, accompanied by the shortening of the extraction time when compared to the reflux-assisted extraction pretreatment.

In conclusion, we have demonstrated the valuable effects of ultrasound and reflux extraction hydrodistillation pretreatments on the yield of essential oils of sage, rosemary, and bay laurel. The benefit of ultrasound is short extraction time, while reflux extraction is performed at mild temperature. Thus, both pretreatments are considered as gentle extraction procedures, resulting in unchanged overall oil quality, as demonstrated by essential oil composition analyzed here. Also, our results point out that when assessing the effect of enzymes as a hydrodistillation pretreatment, proper experiment controls are necessary. In our case, the soaking of plant material in the water at 40 °C was enough to disrupt the cells and additional enzyme activity was redundant.

## 4. Materials and Methods

### 4.1. Plant Material

Fresh leaves were collected from wild plants of rosemary (*Rosmarinus officinalis* L.), sage (*Salvia officinalis* L.), and bay laurel (*Laurus nobilis* L.) in the south Mediterranean region of Croatia in August 2018. The leaves were air-dried at room temperature (20 ± 2 °C) for one week. Dry leaves were packed in polyethylene bags and kept in a dark, dry, and cool place. Before being used for the extractions, the plant material was milled using a house blender (Tefal).

### 4.2. Chemicals

The following chemicals were used: 3,5-dinitrosalicylic acid (DNSA) (98%, Thermo Fisher Scientific, Maharahstra, India), sodium sulphite (Lach-ner, Brno, Chez Republic), sodium hydroxide (Lach-ner, Brno, Chez Republic), phenol (99 + %, Thermo Fisher Scientific, Maharahstra, India), potassium sodium tartrate (Sigma-Aldrich, Buchs, Switzerland), Folin–Ciocalteu reagent (Sigma-Aldrich, Buchs, Switzerland), anhydrous sodium carbonate (Lach-ner, Brno, Chez Republic), gallic acid (anhydrous) for synthesis (Merck, Darmstadt, Germany), cellulase (from *Aspergillus niger*) (Sigma-Aldrich, Tokyo, Japan), pectinase (from *Aspergillus niger*) (Sigma-Aldrich, Buchs, Switzerland), xylanase (from *Theryomyces*, expressed in *Aspergillus oryzae*) (Sigma-Aldrich, Søborg, Denmark), beechwood xylan (Biosynth, Berkshir, England, UK), pectin from citrus peel (74.0%, Sigma-Aldrich, Buchs, Switzerland), sodium carboxymethyl cellulose (Sigma-Aldrich, Buchs, Switzerland), xylose (99%, Sigma-Aldrich, Buchs, Switzerland, D-( + )-glucose (99.5%, Sigma-Aldrich, Buchs, Switzerland), D-( + )-galacturonic acid monohydrate (97.0%, Sigma Aldrich, Buchs, Switzerland), C_9_-C_25_ alkanes, deuterated chloroform for NMR spectroscopy (CDCl_3_-d with 0.03% *v/v* TMS, 99.80%, Eurisotop, Saint-Aubin, France).

### 4.3. 3,5-Dinitrosalicylic Acid (DNSA) Assay for the Determination of Enzyme Activity

The determination of enzyme activity was done according to a DNSA colorimetric method as described in Ghose [42] and Miller [43] with modifications. Enzymes (cellulase, pectinase, and xylanase) were added to MiliQ water to a final concentration of 0.02 mg/mL. Next, 10 mg of the appropriate substrate (cellulose, pectin, and xylan, respectively) was added to 1 mL of enzyme solution and the reaction was incubated at 40 °C for 60 min in a thermoshaker (Biosan, TS-100) at 900 rpm. The sample was then cleared by centrifugation (5 min, 1568× *g*). Next, sample aliquots (600 μL final volume) were mixed with 600 μL of DNSA reagent containing 10.0 g/L DNSA, 0.5 g/L sodium sulfite, 10 g/L sodium hydroxide, and 2 mL/L phenol. The mixtures were incubated for 15 min at 95 °C before adding 200 μL of 40 g/L potassium sodium tartrate solution (1.4 mL final volume). The samples were chilled on ice for 5 min, and then the absorbance at 575 nm was measured (Cary Series UV-Vis Spectrophotometer, Agilent Technologies). Product concentrations were calculated from calibration curves generated with the corresponding reducing sugars (glucose, galacturonic acid, xylose, respectively). If needed, the original samples were appropriately diluted to produce the absorption values within the range of the calibration curves, and dilutions were taken into account in the calculation of the enzymatic activity. One unit of enzymatic activity was defined as the amount of enzyme that releases 1 μmol of reducing sugar per minute under the specified assay conditions.

### 4.4. Determination of Total Phenols for Optimization of Ultrasonic Parameters

The total phenols of plant extracts were determined using the Folin–Ciocalteu method, according to the procedure of Singleton and Rossi [44]. Mass fraction of total phenols was expressed as mg of gallic acid equivalent (GAE) per g dry plant.

### 4.5. Extraction Procedures

Hydrodistillation (HD) was preceded by different pretreatments: hydrodistillation with reflux extraction pretreatment (HD-RE), hydrodistillation with reflux extraction pretreatment assisted with enzymes (pectinase, HD-REP; cellulase, HD-REC; xylanase, HD-REX; pectinase/cellulase/xylanase, HD-REPCX), and hydrodistillation with ultrasonic pretreatment (HD-U). Additionally, HD without pretreatment was performed as a negative control.

For each protocol, 20 g of milled plant material was mixed with 250 mL MiliQ water. For HD-RE, this mixture was subjected to reflux extraction at 40 °C for 1 h under stirring conditions. For enzyme-assisted pretreatments, we have followed previously described protocols [2,19] with slight modifications. Namely, the plant material in MiliQ water was subjected to reflux extraction at 40 °C for 1 h under stirring conditions in the presence of 10 mg pectinase, cellulase, or xylanase, that is, 0.5 mg of enzyme per g of dried plant material, or their combination (5 mg of each enzyme, i.e., 0.75 mg of total enzymes per g of dried plant material). For HD-U, the plant material in MiliQ water was treated with a 14 mm diameter ultrasonic probe (ultrasonic device UP200Ht, Hielscher, Teltow, Germany), at 30% of the maximal ultrasonic power for 10 min. Before the ultrasound extraction, optimal duration, and amplitude were determined (Table 1).

Next, the plant/water mixture was subjected to hydrodistillation using a Clevenger apparatus for 2.5 h. Every extraction was done once. The essential oil was stored in dark glass vials at 4 °C until further analyses.

### 4.6. Gas Chromatograph/Mass Spectrometer (GC-MS) Analysis

GC-MS analyses were carried out with an Agilent Technologies (Palo Alto, CA, USA) gas chromatograph model 7890 A equipped with a mass selective detector (MSD) model 5975 C (Agilent Technologies, Palo Alto, CA, USA) and an HP-5MS 5% phenyl-methylpolysiloxane capillary column (30 m × 0.25 mm, 0.25 μm film thickness, Agilent Technologies, Palo Alto, CA, USA). In brief, the injector and detector temperatures were 250 °C and 300 °C, respectively; the column temperature was held at 70 °C for 2 min, and was then increased from 70 to 200 °C at 3 °C/min, and was finally held at 200 °C for 18 min; 1.0 μL of the sample (10 µL of the oil dissolved in 1 mL of pentane) was injected using split mode (split ratio 1:50). Helium was used as carrier gas (1.0 mL/min). The MSD (EI mode) was operated at 70 eV, the ion source temperature was 230 °C, and the scan range was set to 30–350 amu. Identification of volatile constituents was based on the comparison of their retention indices (RIs), determined relative to the retention times of a homologous series of n-alkanes (C_9_–C_25_), with those reported in the literature and their mass spectra with those of authentic compounds available in our laboratories or those listed in the NIST 08 and Wiley 9 mass spectral libraries (matches higher than 90%). Relative concentrations of components were calculated by the area normalization method without considering response factors.

### 4.7. Statistical Analyses

All variables were log-transformed (using base 10 logs) to improve the data distribution and homogeneity of variances. The data were tested for normality by comparing histograms of the sample data to a normal probability curve, after which the null hypothesis (that data were normally distributed) was rejected. To compare and plot the effects of different pretreatments on the chemical composition of essential oils for each plant (as determined by GC-MS), we applied Spearman’s nonparametric measure of rank correlation using program R v. 3.2.0 [45]. The level of significance was set at *p* ≤ 0.05.

## Figures and Tables

**Figure 1 molecules-25-04818-f001:**
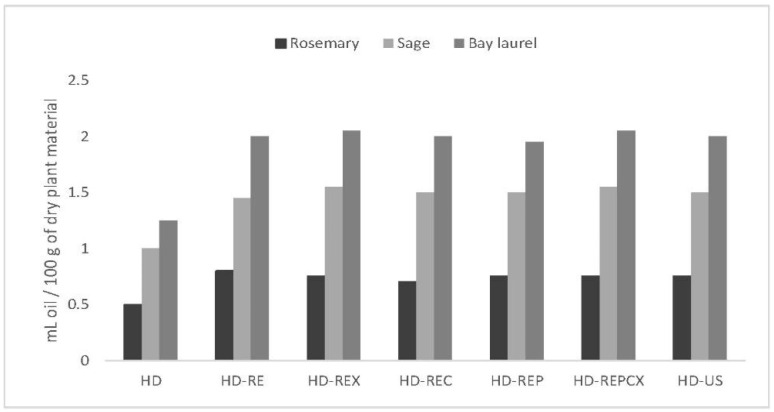
Effect of different hydrodistillation pretreatments on the extraction yield of rosemary, sage, and bay laurel essential oil. Hydrodistillation without pretreatment (negative control)—HD; hydrodistillation with reflux extraction pretreatment—HD-RE; hydrodistillation with reflux extraction pretreatment assisted with enzymes: xylanase—HD-REX; cellulase—HD-REC; pectinase—HD-REP; pectinase + cellulase + xylanase—HD-REPCX; hydrodistillation with ultrasonic pretreatment—HD-US.

**Figure 2 molecules-25-04818-f002:**
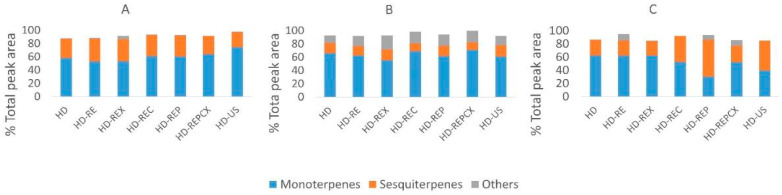
Overview of the chemical composition of sage (**A**), bay laurel (**B**), and rosemary (**C**) essential oils after different hydrodistillation pretreatments, as determined by GC-MS. Hydrodistillation without pretreatment (negative control)—HD; hydrodistillation with reflux extraction pretreatment—HD-RE; hydrodistillation with reflux extraction pretreatment assisted with enzymes: xylanase—HD-REX; cellulase—HD-REC; pectinase—HD-REP; pectinase + cellulase + xylanase—HD-REPCX; hydrodistillation with ultrasonic pretreatment—HD-US.

**Table 1 molecules-25-04818-t001:** Optimization of ultrasound extraction parameters based on the total phenol yield.

	Time	Total Phenol Yield (mg/g)		
		30% of Max. Ultrasonic Power	60% of Max. Ultrasonic Power	90% of Max. Ultrasonic Power
Bay laurel	5 min	16.87 ± 0.18	8.86 ± 0.05	7.32 ± 0.14
	10 min	17.37 ± 0.53	8.27 ± 0.21	5.99 ± 0.00
	15 min	16.86 ± 0.18	8.30 ± 0.21	6.38 ± 0.15
Sage	5 min	29.46 ± 3.14	32.95 ± 2.6	32.20 ± 2.9
	10 min	35.71 ± 5.6	34.24 ± 7.2	36.83 ± 3.9
	15 min	31.25 ± 5.4	31.76 ± 3.4	31.82 ± 0.32
Rosemary	5 min	93.98 ± 1.41	46.35 ± 1.06	50.16 ± 2.75
	10 min	97.44 ± 2.12	45.27 ± 0.07	51.22 ± 0.78
	15 min	103.44 ± 2.12	43.08 ± 0.14	43.12 ± 0.14

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
