# Peer review of "Effect of Enzymatic, Ultrasound, and Reflux Extraction Pretreatments on the Yield and Chemical Composition of Essential Oils"

_molecules, 2020, doi:10.3390/molecules25204818_

Round 1
Reviewer 1 Report
The manuscript by Milijanovic, et al. presents a study that compares the effect of different pre-treatments on sage, bay laurel and rosemary for extracting essential oils in increased yields while maintaining similar compositions. The authors have used GC-MS and NMR to identify a range of compounds to great effect and have placed controls which allow for a better understanding of the effect of pre-treatments. The main conclusion of this work is that enzymatic pre-treatment does not affect the yield and composition of the essential oil produced and that simply soaking the substrate or sonicating provide the same outcome regardless of the presence of enzymes. This conclusion is contradictory to several other studies. In its current state, I do not think the results in this manuscript strongly support this conclusion, at best it suggests that enzymes have minimal effect. Therefore, I recommend that the manuscript only get accepted after major changes.
The authors lack replication in their results which prevents any comparisons to be made accurately. The results in Figures 1 and 2 should be presented in triplicate to make accurate statistically comparisons. The authors state that their statistical analysis shows that the different pre-treatments give essential oils without any significant differences in their chemical composition. In the case of rosemary, Figures 2 and S20 clearly show differences in the composition between treatments. The authors should perform their experiments in triplicate and provide additional statistical analysis (eg: ANOVA or principle component analysis, etc.) to confirm the lack of significant difference in both yield and composition that they claim. The data in its current form can only suggest that enzymes have a minute effect on composition and yield not that they have no effect as stated throughout the manuscript.
Minor comments:
- I think table 1 would better in the supplementary material as it is summarised in Figure 2 already. It is stated that some compounds were confirmed by NMR and by “mass spectra with those of authentic compounds” (line 303). Can this information be added to table 1 with another column or with superscripts to show how strongly some compounds were identified? Additionally, can a comment about the quality of the matches to the MS libraries be added to the experimental section or to the caption of the table.
- Please provide chemical purity and country of origin in 4.2
- Line 264 please provide the xgravity
- Please clarify this statement “Next, diluted sample aliquots (600 μL final volume) resulting in absorption values within the range of the calibration standard curves (glucose, galacturonic acid, xylose, respectively)” lines 264-265
- The term ‘purified water’ is used. Is this RO water, distilled, MilliQ? Please specify.
- Line 288 “Prior the ultrasound extraction, optimal duration and amplitude were determined” Why can these values not be mentioned?
- For the GC-MS analysis please provide the column supplier, how the samples were prepared (eg filtered, diluted) and the MS temperatures.
- For the NMR in supplementary material please provide the number of scans and repetition delay for the 1D experiments.
Reviewer 2 Report
The authors study the effect of different hydrodistillation pretreatments on the yield and chemical composition of essential oils of Rosmarinus officinalis L., Salvia officinalis L. and Laurus nobilis L. The scientific concept of the paper and the experiments are good. This study is interesting to be published; the manuscript adds important information to the existing literature. However, the manuscript needs minor revision before considered for publishing:
* For extractions experiments, How many samples were done? please add.
* The authors must compare the results with those of the literature for example:European Journal of Integrative Medicine (2020) https://doi.org/10.1016/j.eujim.2020.101192 ;TrAC Trends in Analytical Chemistry (2019) https://doi.org/10.1016/j.trac.2019.05.040 ; Industrial Crops and Products (2020) https://doi.org/10.1016/j.indcrop.2020.112094 ;Industrial Crops and Products (2018) https://doi.org/10.1016/j.indcrop.2018.03.021 ;Journal of King Saud University – Science (2019) https://doi.org/10.1016/j.jksus.2018.11.008 ;Saudi Journal of Biological Sciences (2019) https://doi.org/10.1016/j.sjbs.2018.04.008 ;Industrial Crops and Products (2019) https://doi.org/10.1016/j.indcrop.2018.11.006
Author Response
REVIEWER #2
For extractions experiments, How many samples were done? please add.
Every extraction was done once and this information is now included in the manuscript (line 388 of the revised manuscript).
The authors must compare the results with those of the literature for example:European Journal of Integrative Medicine (2020) https://doi.org/10.1016/j.eujim.2020.101192 ;TrAC Trends in Analytical Chemistry (2019) https://doi.org/10.1016/j.trac.2019.05.040 ; Industrial Crops and Products (2020) https://doi.org/10.1016/j.indcrop.2020.112094 ;Industrial Crops and Products (2018) https://doi.org/10.1016/j.indcrop.2018.03.021 ;Journal of King Saud University – Science (2019) https://doi.org/10.1016/j.jksus.2018.11.008 ;Saudi Journal of Biological Sciences (2019) https://doi.org/10.1016/j.sjbs.2018.04.008 ;Industrial Crops and Products (2019) https://doi.org/10.1016/j.indcrop.2018.11.006.
We are thankful to the Reviewer for pointing us to these studies. We cited them in following sentences of the Discussion:
Rosemary and sage are aromatic, medicinal plants that have received particular attention in Lamiaceae family due to their aromatic and chemical composition [2,27,28], while bay laurel is a valuable medicinal plant from Lauraceae family, which has been widely used as spice and flavouring agent [19]. Due to the numerous bioactive compounds, essential oils of these plants show a wide range of biological activities such as antimicrobial, preservative, antioxidant and antifungal, which makes them valuable in a range of applications – from medicinal to food industry [29-33]. Considering these valuable properties, improvement of current and the development of novel essential oils extraction procedures is a subject of many recent studies [28,34,35]. In this study, we have obtained similar volatile profiles of sage, rosemary and bay laurel essential oils as reported elsewhere [2,3,19,35-39]. (lines 180 to 190 of the revised manuscript)
- Ali, A.; Chua, B.L.; Chow, Y.H. An insight into the extraction and fractionation technologies of the essential oils and bioactive compounds in Rosmarinus officinalis L.: Past, present and future. TrAC - Trends Anal. Chem. 2019, 118, 338-351, doi:10.1016/j.trac.2019.05.040.
- Fernández, N.J.; Damiani, N.; Podaza, E.A.; Martucci, J.F.; Fasce, D.; Quiroz, F.; Meretta, P.E.; Quintana, S.; Eguaras, M.J.; Gende, L.B. Laurus nobilis L. Extracts against Paenibacillus larvae: Antimicrobial activity, antioxidant capacity, hygienic behavior and colony strength. Saudi J. Biol. Sci. 2019, doi:10.1016/j.sjbs.2018.04.008.
- Dammak, I.; Hamdi, Z.; Kammoun El Euch, S.; Zemni, H.; Mliki, A.; Hassouna, M.; Lasram, S. Evaluation of antifungal and anti-ochratoxigenic activities of Salvia officinalis, Lavandula dentata and Laurus nobilis essential oils and a major monoterpene constituent 1,8-cineole against Aspergillus carbonarius. Ind. Crops Prod. 2019, 128, 85-93. doi:10.1016/j.indcrop.2018.11.006.
- Ferreira, D.F.; Lucas, B.N.; Voss, M.; Santos, D.; Mello, P.A.; Wagner, R.; Cravotto, G.; Barin, J.S. Solvent-free simultaneous extraction of volatile and non-volatile antioxidants from rosemary (Rosmarinus officinalis L.) by microwave hydrodiffusion and gravity. Ind. Crops Prod. 2020, 145, 112094, doi:10.1016/j.indcrop.2020.112094.
- Al Zuhairi, J.J.M.J.; Jookar Kashi, F.; Rahimi-Moghaddam, A.; Yazdani, M. Antioxidant, cytotoxic and antibacterial activity of Rosmarinus officinalis L. essential oil against bacteria isolated from urinary tract infection. Eur. J. Integr. Med. 2020, 38, 101192, doi:10.1016/j.eujim.2020.101192.
- Vosoughi, N.; Gomarian, M.; Ghasemi Pirbalouti, A.; Khaghani, S.; Malekpoor, F. Essential oil composition and total phenolic, flavonoid contents, and antioxidant activity of sage (Salvia officinalis L.) extract under chitosan application and irrigation frequencies. Ind. Crops Prod. 2018, 117, 366-374, doi:10.1016/j.indcrop.2018.03.021.
- Khiva, Z.; Havani, M.; Garnar, A.; Kharchouf, S.; Amine, S.; Berrekhis, F.; Bouzoubae, A.; Zair, T.; Elhilali, F. Valorization of the Salvia officinalis L of the Morocco bioactive extracts: Phytochemistry, antioxidant and anticorrosive activities. J. King Saud Univ. Sci. 2019, doi:10.1016/j.jksus.2018.11.008
Round 2
Reviewer 1 Report
The revised manuscript by Milijanovic, et al. has undergone significant improvements and the authors have clearly addressed all comments made previously. The author’s claim of minimal changes in essential oil yield and composition with and without enzymatic pre-treatment are supported by the results and are of interest to the readership of Molecules.
I have 2 minor comments:
First, I understand the authors not wanting to present redundant data but I think adding the ANOVA table to the supplementary material would show readers that the statistical analysis has been confirmed. However, this is not needed for publication.
Secondly, in the caption of figure 1 please state that the results are from triplicate measurements